# Radiogenomics Analysis Linking Multiparametric MRI and Transcriptomics in Prostate Cancer

**DOI:** 10.3390/cancers15123074

**Published:** 2023-06-06

**Authors:** Catarina Dinis Fernandes, Annekoos Schaap, Joan Kant, Petra van Houdt, Hessel Wijkstra, Elise Bekers, Simon Linder, Andries M. Bergman, Uulke van der Heide, Massimo Mischi, Wilbert Zwart, Federica Eduati, Simona Turco

**Affiliations:** 1Electrical Engineering Department, Eindhoven University of Technology, 5600 MB Eindhoven, The Netherlandsm.mischi@tue.nl (M.M.); 2Biomedical Engineering—Computational Biology Department, Eindhoven University of Technology, 5612 AZ Eindhoven, The Netherlandsw.zwart@nki.nl (W.Z.); f.eduati@tue.nl (F.E.); 3Department of Radiation Oncology, The Netherlands Cancer Institute, 1066 CX Amsterdam, The Netherlands; 4Department of Urology, Amsterdam University Medical Centers, 1100 DD Amsterdam, The Netherlands; 5Department of Pathology, The Netherlands Cancer Institute, 1066 CX Amsterdam, The Netherlands; 6Division of Oncogenomics, The Netherlands Cancer Institute, 1066 CX Amsterdam, The Netherlands; 7Division of Medical Oncology, The Netherlands Cancer Institute, 1066 CX Amsterdam, The Netherlands; 8Institute for Complex Molecular Systems, Eindhoven University of Technology, 5600 MB Eindhoven, The Netherlands

**Keywords:** magnetic resonance imaging, prostate cancer, radiogenomics, machine learning

## Abstract

**Simple Summary:**

Prostate cancer is a global health burden. Multi-parametric magnetic resonance imaging is the recommended imaging modality for diagnosis. The recommended treatment differs based on tumor aggressiveness, typically assessed with the use of invasive techniques such as tumor biopsies. By studying the relationship between imaging characteristics and the genomic information obtained from tumor biopsies, it might be possible to detect aggressive tumor characteristics based solely on imaging, which could eventually be used to non-invasively inform on patient-tailored treatments. In this study, we extracted a large number of imaging features and found significant correlations between them and the aggressiveness of the tumor. We additionally investigated transcriptomic features (i.e., patterns of gene expression) associated with tumor aggressiveness and identified significant correlations with perfusion-related image features, highlighting a link between what is visible on the diagnostic images and the underlying genomic information of the tumors.

**Abstract:**

Prostate cancer (PCa) is a highly prevalent cancer type with a heterogeneous prognosis. An accurate assessment of tumor aggressiveness can pave the way for tailored treatment strategies, potentially leading to better outcomes. While tumor aggressiveness is typically assessed based on invasive methods (e.g., biopsy), radiogenomics, combining diagnostic imaging with genomic information can help uncover aggressive (imaging) phenotypes, which in turn can provide non-invasive advice on individualized treatment regimens. In this study, we carried out a parallel analysis on both imaging and transcriptomics data in order to identify features associated with clinically significant PCa (defined as an ISUP grade ≥ 3), subsequently evaluating the correlation between them. Textural imaging features were extracted from multi-parametric MRI sequences (T2W, DWI, and DCE) and combined with DCE-derived parametric pharmacokinetic maps obtained using magnetic resonance dispersion imaging (MRDI). A transcriptomic analysis was performed to derive functional features on transcription factors (TFs), and pathway activity from RNA sequencing data, here referred to as transcriptomic features. For both the imaging and transcriptomic features, different machine learning models were separately trained and optimized to classify tumors in either clinically insignificant or significant PCa. These models were validated in an independent cohort and model performance was used to isolate a subset of relevant imaging and transcriptomic features to be further investigated. A final set of 31 imaging features was correlated to 33 transcriptomic features obtained on the same tumors. Five significant correlations (*p* < 0.05) were found, of which, three had moderate strength (|r| ≥ 0.5). The strongest significant correlations were seen between a perfusion-based imaging feature—MRDI *A* median—and the activities of the TFs STAT6 (−0.64) and TFAP2A (−0.50). A higher-order T2W textural feature was also significantly correlated to the activity of the TF STAT6 (−0.58). STAT6 plays an important role in controlling cell proliferation and migration. Loss of the AP2alpha protein expression, quantified by TFAP2A, has been strongly associated with aggressiveness and progression in PCa. According to our findings, a combination of texture features extracted from T2W and DCE, as well as perfusion-based pharmacokinetic features, can be considered for the prediction of clinically significant PCa, with the pharmacokinetic MRDI *A* feature being the most correlated with the underlying transcriptomic information. These results highlight a link between quantitative imaging features and the underlying transcriptomic landscape of prostate tumors.

## 1. Introduction

Prostate cancer (PCa) has the highest incidence among men worldwide, accounting for one-third of all cancer diagnoses in males [1]. In 2020, with 375,000 deaths worldwide, PCa was the fifth leading cause of cancer-related death among men, taking the lead in countries located in sub-Saharan Africa, the Caribbean, and Central and South America [2]. In the traditional diagnostic pathway of PCa, patients with suspicious findings on serum prostate-specific antigen (PSA) testing and/or digital rectal examination (DRE) are typically referred for biopsy, with or without prior imaging [3]. The risk for biochemical recurrence is then calculated based on PSA values at diagnosis, the Gleason score at biopsy, and the clinical stage [3]. While active surveillance is recommended for low-risk disease, radical treatment (radical prostatectomy or radiotherapy) is offered to intermediate-risk patients, and multimodal therapeutic approaches, combining active treatment and systemic therapies, should be considered for advanced, high-risk PCa [4]. This pathway has been shown to lead to both underdiagnosis and misdiagnosis of clinically significant PCa, as well as overdiagnosis and overtreatment of insignificant PCa [4,5,6].

Spurring from a need for more accurate diagnostic and prognostic markers, genomic-based tests have been proposed and validated [7,8]. However, the costs and incomplete evidence for long-term outcomes have hampered the clinical implementation of genomic tests [9]; currently, they are only advised when risk upgrading may change clinical management [9] and for advanced PCa [10]. Additionally, as genomic tests are performed on biopsied samples, they bear the same risks (infection and bleeding) and discomfort associated with biopsies [11].

Multi-parametric magnetic resonance (mpMRI), combining anatomical T2-weighted imaging, with functional diffusion-weighted imaging (DWI) and dynamic-contrast enhanced (DCE) imaging [5,12], is an established non-invasive tool for PCa imaging with the potential to improve PCa diagnosis and management [4,5,6,13]. According to the latest updates from the American and European guidelines [3,4,9,14], mpMRI is recommended for early detection in patients with suspicious PSA and/or DRE findings, for targeting biopsies, for augmenting risk-stratification in active surveillance patients, and for local tumor staging. However, despite improvements achieved through the updated prostate Prostate Imaging–Reporting and Data System (PI-RADS) scoring [15], low specificity and inter-observer variability remain concerns [4,5,14].

In the past decades, advanced computational approaches for quantitative mpMRI analyses have shown promise for (semi)automatic extraction of imaging features, possibly improving diagnostic accuracy, while mitigating the unavoidable variability inherent in human-based interpretation and scoring systems [5,12,16,17]. While the calculation of the apparent diffusion coefficient (ADC) from DWI [18] and the pharmacokinetic analysis of the DCE sequence [19,20] are well-established quantification methods for prostate mpMRI, radiomic analysis has recently emerged for the high-throughput extraction of imaging features, ranging from size, shape, and volume, to statistical properties of the image intensity, and more complex fractal, texture, and transform-based features [21,22,23,24]. Currently, radiomic features typically require the identification and manual delineation of a region of interest, such as a tumor lesion. From this region, one value of each feature is extracted, thus losing the granularity available in ADC and pharmacokinetic maps, in which, instead, the parameters are extracted at each voxel.

Aligned with the current paradigm shift towards precision oncology, the concept of radiogenomics has more recently emerged, aimed at linking radiomic features with the underlying molecular and genomic characteristics of tumor tissues and micro-environments [21,24,25]. The overarching goal is to identify radiomic features that correlate with and complement genomic-based markers for improved diagnosis, risk stratification, and therapeutic decision-making. Although the word “genomics” technically refers to the presence of genes, in the context of radiogenomics, it is also often used to refer to gene expression levels (transcriptomics).

In the prostate, initial studies on radiogenomics focused on the association of the PCA3 score, a urine marker for aggressive prostate cancer, with a qualitative assessment of mpMRI by PI-RADS or Likert scoring [26,27,28], with conflicting results. While a significant association was found between the PCA3 score and PI-RADS [28], PCA3 was not associated with cancer detection for high-suspicion mpMRI cases [27]. Several studies investigated the association between mpMRI findings (e.g., tumor visibility and PI-RADS) with transcriptomics-based scores and molecular biomarkers from biopsied cancer tissue, including Prolaris, Decipher, the genomic prostate score, and expression of phosphatase and tensin homolog (PTEN) [21,29,30,31,32]. Although transcriptomics scores were generally predictive of adverse pathology results (the Gleason score) [27,29,31,32] and an association was found between PI-RADS and genomic scores [30,31], a wide range of genomic-based score values was observed across all PI-RADS categories [29,30]. Beyond a qualitative assessment of mpMRI, a few studies extracted quantitative features by e.g., radiomics [33,34,35,36,37,38]. Focusing only on PTEN expression, McCann et al. [33] found a significant correlation with $k_{ep}$, a pharmacokinetic parameter extracted from DCE, which is related to vascular permeability. However, this result is in contrast with a later study, in which PTEN expression only correlated with mean ADC values [36]. Investigating tumor hypoxia, Sun et al. [38] found a set of texture features extracted from T2W imaging to be significantly associated with hypoxia-related genes. While previous studies only focused on one or a few genes, broader genes and microRNA expression analyses were performed in [35,37]. Both studies demonstrated that radiomics features are correlated with gene signatures related to immune response activation, cycle cell processes, and DNA repair [35]; a radiomic-based classifier was able to predict Gleason scores and Decipher scores with moderate accuracy [35]. With few exceptions [33,36], however, the DCE series is not included in the quantitative mpMRI analysis; this is possibly due to the hurdles of PK analysis by the conventional Tofts model [19], which requires the separate estimation of an arterial input function. Moreover, only a subset of intensity-based and texture-based features was extracted from the whole tumor ROI.

In this study, we investigate the relationship between imaging and transcriptomic features in aggressive PCa. To this end, we first identify the imaging and transcriptomic features that are most predictive of aggressive PCa by machine learning and then analyze their correlation. In contrast to previous studies, we investigate a larger set of texture features on all mpMRI imaging sequences and we adopt a moving-window strategy to extract features at each imaging voxel, allowing the investigation of the feature spatial distribution and heterogeneity in the tumor ROI. Additionally, we perform magnetic resonance dispersion imaging (MRDI) on the DCE sequence for automatic estimation of PK parameters related to microvascular perfusion, architecture, and permeability, without the need for an arterial input function [20,39].

## 2. Materials and Methods

### 2.1. Data Collection

In this work, the genomic model was based on transcriptomic features, while the radiomics model was based on imaging features extracted from mpMRI. As described in Figure 1, three independent datasets were used in this study. For both genomics and imaging, separate training and test datasets were used; the training dataset was used for the optimization of machine learning model(s) and feature selection, while the test dataset was used for performance evaluation and for investigating the correlation between the most relevant imaging and transcriptomic features.

The genomics training dataset consisted of prostate adenocarcinoma patients from the TCGA dataset. Bulk RNA-seq data were obtained through the online tool from the Broad Institute (https://gdac.broadinstitute.org/, version released 28 January 2016), and clinical information was downloaded from the GDC Data Portal (https://portal.gdc.cancer.gov/, accessed on 18 September 2022). We used a subset of 331 patients passing quality control and with revised clinical data, as described in [40].

The imaging training dataset was retrospectively collected from 3 different Dutch institutes (Amsterdam University Medical Center, The Netherlands Cancer Institute, and Radboud University Medical Center), as part of the Prostate Cancer Molecular Medicine study. Patients were eligible for inclusion if they underwent radical prostatectomy due to biopsy-proven PCa and if a preoperative mpMRI, including T2W, DWI, and DCE sequences, was available. Reasons for exclusion were: the presence of (movement) artifacts in one of the sequences, the temporal resolution of the DCE sequence was not sufficiently high for the MRDI analysis, and missing histopathology analysis. According to the inclusion criteria, a total of 45 patients were included.

For both imaging and genomics, the test dataset consisted of PCa patients treated at The Netherlands Cancer Institute (Amsterdam, the Netherlands). The genomics test dataset included 91 patients, from whom bulk RNA-seq was taken from a single identified lesion on the biopsy sample. In the sporadic case when multiple lesions were visible, the RNA sample was taken from the index lesion. A subset of patients was also imaged with mpMRI and biopsied prior to radical prostatectomy, which constituted the imaging test dataset. The inclusion criteria for the imaging test dataset were the same as for the imaging training dataset, with the addition of the RNA sample being available, resulting in 35 patients being included.

The histopathological analysis of the prostate after radical prostatectomy was available for each patient included in the study. The pathologically derived ISUP grade [41] was then used to stratify patients into clinically insignificant (ISUP ≤2) and significant (ISUP ≥3) PCa. An overview of the distributions of ISUP grades for all patients included in this study is provided in Table 1.

In the imaging training dataset, prostate specimens were first fixated in formalin and then cut into 4 mm slices and photographed. The microscopic analysis was then performed by a pathologist to mark cancer regions on the basis of cellular differentiation. In the test dataset, the surgically removed prostate was stained with hematoxylin and eosin (H&E), and a pathologist delineated the tumor in the available pathology slides.

#### 2.1.1. Imaging Data Pre-Processing

The MRI acquisition settings are summarized in the Appendix A for the training and test datasets, respectively. While for the training dataset, ADC maps automatically generated by the scanner were available. For the test dataset, the ADC maps were often missing and/or of poor quality; therefore, new maps were generated for the whole test dataset. A minimum of two *b* values between 100 and 1000 s/mm^2^ were used to calculate the maps. As described in the PI-RADS guidelines [42], the ADC maps were generated using the conventional monoexponential model, where the signal intensity on DWI has a log-linear relationship with the *b* values. The best-fitting linear equation was found using a least squares approximation, with the slope yielding the ADC value.

Prostate zonal segmentation was performed on all patients using a modified multichannel U-net neural network to extract the contours of the whole prostate, the central gland, and the peripheral zone [43]. The obtained contours were reviewed and adjusted manually when necessary. For both datasets, the three imaging sequences were resampled onto the T2W imaging grid for each patient.

For both training and test datasets, tumor regions of interest (ROIs) were defined based on the histopathology contours. In the training dataset, tumor delineations were available for each histopathology slide, which were used to guide visual matching and manual delineation of a tumor ROI on the MRI scans using ITK snap [44]. In the test dataset, one to two H&E-stained slides were available per patient. These were then scanned and deformably registered to the T2W grid. Firstly, the T2W slice that best matched the H&E stained slide was identified, and then anatomical landmarks were marked on both images (Figure 2a,b). The H&E samples were deformably registered to the grid of T2W MRI by using a coherent point drift algorithm, based on the manually selected matching landmarks on both images. The transformed H&E slide (Figure 2c) was then used to draw and propagate the tumor delineations onto the T2W slice (Figure 2d). The tumor ROI was delineated using ITK snap [44].

In order to account for registration errors in defining the tumor ROIs, the registration error was derived per patient by measuring the distance between matching visual landmarks on both images. The tumor ROIs were then eroded with an error margin of half the average registration error. The size of the error was determined to be 1.19 mm. Therefore, the tumor contours were eroded with a margin of 0.594 mm (approximately 2 pixels in the T2W grid).

For training, the full extent of the identified tumor lesions—often comprising several MRI slices after matching to the corresponding pathology slides—were included. In the sporadic case when there were multiple lesions, the GS of the index lesion was assigned. In the test dataset, for each patient, one H&E slide was successfully registered to a corresponding MRI slice. Additionally, as the transcriptomics information was only available for the index lesion, the analysis of the test dataset focused on the index tumor ROI as identified in the single MRI slice with histopathology matching. This resulted in a total of 218 and 35 lesion ROIs for model training and testing, respectively.

### 2.2. Transcriptomics Analysis Pipeline

#### 2.2.1. Feature Extraction

To summarize the transcript expression levels, the counts and transcript per million (TPM) were computed from the bulk RNA sequencing data by using the first two steps of the pipeline from quanTIseq [45], where RNA sequencing reads were pre-processed with Trimmomatic [46]; consequently, the raw counts and TPM were quantified with Kallisto [47]. The RNA sequencing data were processed using the R package EaSIeR [48], to derive quantitative features describing different facets of the tumor microenvironment, such as the activity of pathways and transcription factors (TFs). In particular, we made use of the PROGENy tool [49] to compute the activity of 14 intracellular signaling pathways based on the expressions of genes that are differentially regulated during pathway-specific perturbation experiments. The DoRothEA tool [50] was used to infer the activity of 118 TFs based on the expressions of their target genes. In addition, the gene signature scores for PORTOS and Decipher were derived. PORTOS [51] and Decipher [52] are two well-established PCa signatures, the first is used to predict the response to postoperative radiotherapy and the latter to predict the response to radical prostatectomy. The model for the PORTOS score was developed for microarray data, and the model for Decipher was proprietary, as a result, no scores could be computed. However, we explored three alternative methods to compute scores based on the expressions of the signature genes: ssGSEA [53], geometric mean [54], and PCA [55]. For the PCA method, we use the values from the first principal component as the scores. In this way, 6 signature scores were obtained (3 from PORTOS and 3 from Decipher), leading to a total of 138 transcriptomic features. Appendix A, included in the Appendix A, lists the included features.

#### 2.2.2. Machine Learning for Feature Selection

A regularized logistic regression model was optimized on the transcriptomic training dataset by a nested stratified cross-validation approach with 25 repeats, 10 outer loops, and 5 inner loops. In the nested approach, the inner loop is used to optimize the hyperparameters, while the outer loop is used for performance evaluation and model selection. In both loops, folds were stratified to ensure that the folds were balanced between the two classes (insignificant vs. significant PCa), and features were standardized in each outer loop. The models were optimized by minimizing the log loss and penalized with elastic net regularization. Features were ranked by importance by averaging the weight coefficients obtained for each feature at each repetition of the cross-validation procedure and converting them to odds ratios (ORs) using $\mathrm{OR}=e^{w}$, with *w* being the coefficient of a feature. The OR quantifies the relative odds of having clinically significant PCa given the value of a feature, where an OR > 1 means that a higher value for this feature is associated with higher odds of clinically significant PCa and the opposite with an OR < 1 [56]. Features for which the OR was very close to 1, with a margin of 0.1 (|OR−1|<0.1), were excluded from further investigation. This procedure was done separately to perform feature ranking and selection for the transcription factors and pathway activity. For PORTOS/Decipher signature scores, all six features were further investigated.

### 2.3. Image Analysis Pipeline

An overview of the imaging feature extraction and machine learning pipelines is presented in Figure 3. For each imaging slice where a lesion is delineated, texture analysis by a moving window is applied to T2W, ADC, DCE wash-in, and peak images. In parallel, the MRDI analysis is applied to the full DCE-MRI video to obtain MRDI parametric maps. Summary statistics are calculated from the voxels within the lesion’s ROI. Dimensionality reduction is then applied by correlation thresholding and maximum relevance minimum redundancy (mRMR) feature selection [57]. The remaining features were used as input to the machine learning pipeline. For each tested classifier, 20 repetitions of the grouped 5-fold cross-validation were performed on the training data to find the best model, consisting of the best subset of features and the best hyperparameters. The performance of the best model was then evaluated on the test data. All the steps of the pipeline are detailed hereafter.

#### 2.3.1. Feature Extraction

**MRDI analysis**. A pharmacokinetic analysis approach, denoted as magnetic resonance dispersion imaging (MRDI) [20,39], was performed on the DCE-MRI time series in order to derive parameters describing microvascular perfusion, architecture, and permeability. The MRDI analysis consists of a two-compartment model, with one compartment accounting for the intravascular contrast transport, and one compartment accounting for extravasation. While extravasation is modeled as a mono-exponential decay, a convective dispersion model is adopted to describe the contrast agent transport within the (micro)vascular network. A local solution of this model, the modified local density random walk, provides the intravascular contrast concentration, $C_{i}\left(t\right)$. As described in [20,39], this model circumvents the need for arterial input function estimation, and a reduced dispersion model can be used to calculate the PK parameters:(1)$$C\left(t\right)=A\sqrt{\frac{\kappa }{2\pi (t-t_{0})}}e^{\frac{-\kappa {\left(t-t_{0}-\mu \right)}^{2}}{2(t-t_{0})}}*e^{-k_{ep}t},$$
where $A=\alpha K^{trans}$, with α equal to the time integral of the intravascular concentration and $K^{trans}$ the forward volume transfer constant; μ is the intravascular mean transit time; $t_{0}$ is the theoretical injection time; and κ is the dispersion parameter. As a separate estimation of $K^{trans}$ is not possible, the parameter *A* is in turn investigated. Equation (Equation 1) was used to fit the time-intensity curves obtained at each voxel of the DCE sequence, resulting in the PK parameters *A*, μ, $t_{0}$, $k_{ep}$, and κ. Figure 4 shows an example of the extracted *A*, $t_{0}$, and κ maps for one patient. The aforementioned analysis was performed using MATLAB R2022a.

**Texture analysis**. While the texture analysis of T2W and DWI is straightforward, due to the dynamic nature of the DCE sequence, suitable temporal frames first need to be selected. As malignant lesions have been associated with distinguishable enhancement patterns at wash-in and higher peak intensities [58], here, the wash-in and peak frames were isolated and used as input for texture feature extraction. The time frames were chosen by analyzing the average time-intensity curve within the prostate region in a single slice located in the middle of the gland. The frame with the maximum mean intensity was defined as the peak frame. As for the wash-in, this was defined as the frame for which the time-intensity curve displayed the highest positive gradient. The candidate wash-in and peak frames were visually inspected and, when necessary, a more representative frame was chosen.

Using the Vallièrres et al. Matlab toolbox [59], texture feature maps were extracted from T2W, ADC, DCE peak, and DCE wash-in frames (Figure 4). The features were extracted using a moving window of size of 15 × 15 pixels, moved with a stride of 3 pixels. The window size and stride were chosen empirically to prevent the loss of local characteristics of the features while ensuring a meaningful distribution of the intensity values in the window. Compared to the traditional ROI-based feature extraction, this approach allows for the extraction of texture features at each voxel, thus obtaining parametric texture maps (see example in Figure 4). In each window, the intensity was quantized to 64 grayscale levels using equal probability quantization. A total of 43 texture features were extracted for each image, as summarized in Table 2, for a total of 172 texture maps.

#### 2.3.2. Summary Statistics and Feature Filtering

In total, 181 parametric maps were obtained, including 4 raw grayscale images (T2W, DWI, DCE_wash_in, DCE_peak), 43 texture maps obtained for each of the 4 raw images, and 5 MRDI maps. In order to capture the feature distribution in the lesion, the following summary statistics were calculated in the lesion ROI for each feature map: median, 10th and 90th percentiles, standard deviation (std), mean, skewness, and kurtosis. This resulted in a total of 1267 features. Redundant features were removed by correlation analysis, in which for each pair of features with a correlation larger or equal to 0.9, only the feature with the largest correlation to the outcome variable was kept. The principal component analysis (PCA) was then used to inform on the number *m* of components necessary to explain 95% of the variance. The features were ranked using mRMR [57] and the top *m* features were selected as input to the machine learning pipeline, described hereafter.

#### 2.3.3. Machine Learning Pipeline

An overview of the machine learning pipeline is schematically shown in Figure 3. After dimensionality reduction, the remaining *m* features were given as input to a grouped stratified 5-fold cross-validation procedure, which ensured that lesions from the same patient were always grouped in the same fold; in this way, data from the same patient can never be in both the training and validation folds. Additionally, folds were stratified to preserve class proportions. The procedure was repeated 20 times with different splits of the patients in the folds.

At each iteration of the cross-validation loop, feature selection was performed to determine both the best number and subset of features, and the hyperparameters of three machine learning models—logistic regression (LR), support vector machine (SVM), and k-nearest neighbors (kNN)—were optimized. The LR, KNN, and SVM models were chosen as they are widely used, simple to implement, and computationally efficient. The chosen optimization criterion was the area under the ROC curve (AUC), which balances between sensitivity (SENS) and specificity (SPEC). To cope with class imbalance, the “class weight” hyperparameter was set to “balanced” for the LR and SVM; however, this was not possible for the kNN. For all classifiers, feature selection was performed by recursive feature selection, using a basic logistic regression as a classifier. Additionally, to investigate the importance of each feature in the prediction, the permutation feature importance (PFI) was calculated for each optimized classifier. Briefly, the PFI is calculated as the decrease in the model score after randomly permuting the values of one feature, so that the feature loses any relationship with the class. In this way, it is possible to individually ascertain the importance of each feature in a model-agnostic fashion. The calculation was repeated with 10 random permutations, and the mean PFI was then normalized so that the most important feature has a mean PFI equal to 1.

Finally, the optimized models were validated on the test dataset, for which the PFI was also calculated. The performances of the models were evaluated on both training and test datasets by calculating sensitivity, specificity, accuracy (ACC), balanced accuracy (bACC), and AUC.

All steps of the machine learning pipeline were performed using scikit-learn [60].

### 2.4. Correlation with Transcriptomic Features

The selected transcriptomic features were correlated with the most relevant imaging features, i.e., the selected features for the best classifier. The correlations were corrected using false discovery rate correction for multiple testing by the Benjamini–Hochberg method [61], with a family-wise error rate set to 0.2. The correlations were clustered using unsupervised hierarchical clustering with Euclidean distance as a similarity metric with the R package ComplexHeatmap [62]. Only significant correlations (α=0.05) were further investigated.

## 3. Results

### 3.1. Patient Stratification

According to the ISUP-based criterion described in Section 2.1, the transcriptomic training data consisted of 165 clinically insignificant and 166 clinically significant PCa, while in the test dataset, these were 57 and 34, respectively. For the imaging data, the training dataset included 30 clinically insignificant PCa patients, yielding 140 ROIs, and 15 clinically significant PCa patients, yielding 78 ROIs. This resulted in a total of 218 ROIs used to train and optimize the model parameters. For the test imaging data, there were 11 clinically insignificant and 24 clinically significant PCa patients, each providing one ROI, as explained in Section 2.1.1.

### 3.2. Dimensionality Reduction and Feature Selection

First, we performed knowledge-based dimensionality reduction on the transcriptomics data by deriving 132 functional features describing pathways and TF activity from the expressions of more than 20,000 genes, as described in Section 2.2.1. Then the transcriptomic features were selected according to the OR criterion described in Section 2.2.2, resulting in 10 pathways and 17 transcription factors. Together with the 6 PORTOS/Decipher signature scores, a total of 33 transcriptomic features were input to the correlation analysis—Table 3. For the imaging features, removing highly correlated features reduced the feature set from 1267 to 609. A total of 122 principal components was necessary to explain 95% variance within the imaging training dataset, and this number was used to empirically define the subset of features to select for further analysis. Thus, the top-ranking mRMR 122 features were used to further optimize and train the models.

### 3.3. Imaging Classifiers

For each imaging classifier, the performance was evaluated at each repetition of the k-fold cross-validation framework to find the optimal number of features and the best hyperparameters. These are presented for each model in Table 4. The performances of the imaging classifiers on both training and test datasets are presented in Table 5. Both LR and SVM achieve good performances on the training set (AUC = 0.82 and AUC = 0.83, respectively), and moderate performances on the test dataset (AUC = 0.74 and AUC = 0.72, respectively).

### 3.4. Permutation Feature Importance of Imaging Features

The normalized PFIs of the top 10 features obtained on the training dataset for all imaging classifiers are shown in Figure 5. As shown by the coloring, most features were shared by the classifiers. Interestingly, LR and SVM share three of their top four features, of which two are also shared with the kNN. These are the standard deviation of MRDI parameter *A* and the kurtosis of the texture feature GLSZM-LZLGE extracted from T2W. Similar findings are observed in the test set (Figure 6), where several features are common among all classifiers, and the top four features are the same for LR and kNN. Notably, for LR and SVM, the kurtosis of the texture feature GLSZM-LZLGE from T2W and the 90th percentile of the texture feature GLCM-variance from the DCE peak are in the top 10 for both training and test datasets. Additionally, the standard deviation of MRDI parameter *A* is in the top 10 for the SVM for both training and test datasets.

### 3.5. Correlation with Transcriptomic Features

The heatmap of the Pearson correlation coefficient is shown in Figure 7, with significant correlations (adjusted *p*< 0.05) highlighted by an asterisk. There were 5 significant correlations, of which 3 had moderate strength (≥0.5). The strongest significant correlations were seen between the MRDI *A* median and the activity of the transcription factors STAT6 (−0.64) and TFAP2A (−0.50). The T2W GLRLM SRLGE kurtosis textural feature was also significantly correlated to the transcription factor STAT6 (−0.58).

## 4. Discussion

The literature on PCa radiogenomics is scarce [21], and to our knowledge, none has explored the relationship between transcriptomics and quantitative radiomic features extracted from all three mpMRI sequences derived from a multicenter dataset. The identification of radiomic features describing aggressive PCa phenotypes can pave the way for providing non-invasive advice on patient-tailored treatment strategies. Our results show that spatiotemporal and textural features describing PCa anatomy and perfusion characteristics can be (cor)related to the underlying transcriptomic landscape.

The imaging machine’s learning framework employed a moving window approach to extract texture features from T2W, ADC, and relevant time points of the dynamic DCE series. This method defines the ROI as a moving window, enabling the extraction of texture feature values at each imaging pixel, and resulting in a complete parametric map. In this way, it is possible to preserve the spatial characteristics of the features and retain potential texture heterogeneity within the image. In fact, our PFI analysis demonstrated that statistics such as kurtosis and standard deviation, which are related to the spatial distribution of the feature values, are more important than the means and median in the classification of clinically significant cancer, providing support for the adopted strategy. The best radiomic feature classifier was LR, which obtained a fair performance (AUC = 0.74) in predicting clinically significant PCa on an independent dataset. All models resulted in a high specificity (>0.80), but poor sensitivity. From a clinical point of view, this is undesirable, as clinically significant PCa might be missed. This is likely related to the naturally occurring data imbalance in our datasets, which is a result of center-specific practices (i.e., which patients are treated with prostatectomy). Some of the main limitations of this study include the relatively small datasets used for the training of the imaging models, and more importantly, in the fact that they are imbalanced, such that the ratio of patients per group is inverse in the train and test datasets. In the imaging train dataset, 33% of the ROIs are of clinically significant PCa, while for the test dataset, this value is 69%. To cope with the data imbalance, for both LR and SVM classifiers, the class weight was set to “balanced”, which automatically adjusts weights to be inversely proportional to the class frequencies. Such an approach was not possible with KNN, which might explain this classifier’s poor performance. The small sample size is also a limitation considering the optimal number of features to be used and might be responsible for some of the variability observed in terms of performance, as we are currently at the boundary of what can be explored with a dataset of this size. Larger datasets would allow for the validation of the results here presented as well as an investigation of a larger number of features. Additionally, hyperparameter optimization was performed using AUC as an optimization metric, as it leverages both sensitivity and specificity and deals in this way with the imbalance in the datasets. In addition to coping with the imbalanced dataset, this choice is also motivated by the clinical desire to optimize both sensitivity and specificity. At last, we performed a k-fold cross-validation scheme within the training dataset, repeating the procedure for 20 different random splits of the tumors in the 5 folds, so as to reduce the dependence of the estimated performance on the individual random split. Our AUC performance is slightly lower than reported in studies using the same Gleason score-based stratification (AUC∼0.80) [21], but it is important to stress that ours is a multicenter study, likely incorporating larger variabilities in the measurement data. Radiomic features are known to be sensitive to image characteristics [63]; thus, by incorporating data from multiple institutes, scanners, and field strengths, we aimed to identify features that are robust to image variations.

The model-agnostic feature importance analysis using PFI showed that the highest-ranking imaging features overlapped between classifiers. Looking at the top four ranking features, three of them were shared between LR and SVM, and two between LR and kNN. Three features were shared within the top five for all classifiers: MRDI *A* std, T2 GLSZM LZLGE kurtosis, and the DCE wash-in NGTDM strength std, highlighting the importance of texture and spatiotemporal perfusion-based features for the stratification of clinically significant PCa (Figure 5). Despite the differences between our test and train datasets, 4 out of the top 10 highest-ranking radiomic features were common in both the train and test datasets, with T2 GLSZM LZLGE kurtosis appearing again as one of the overlapping features (Figure 6). The GLSZM features are rotation-independent and useful in evaluating non-periodic and heterogeneous textures as they measure consecutive groups of pixels with the same intensity, regardless of the direction. The large zone low gray-level emphasis (LZLGE) feature computes the zone distribution of the upper right quadrant of the GLSZM, with higher values of LZLGE being correlated to a greater number of large zones with low intensities [22]. The kurtosis is the fourth statistical moment of a probability distribution and can be interpreted as a measure of the relative weight of the tails, taking the normal distribution as a reference. Large kurtosis indicates that tail values are more extreme compared to a normal distribution [22]. In our context, a small number of very low or very high values of large groups of pixels with lower intensities might be observed in the investigated tumors. Low T2W signal intensity is a characteristic trait of PCa, and very large regions of low gray-level values could represent larger tumors and more aggressive PCa. MRDI *A* is given by the product of $K^{trans}$ and α, with α being equal to the time integral of the intravascular concentration, and reflecting both extravasation and perfusion [20,39], which are typically increased due to angiogenesis. A higher standard deviation on this parameter’s values represents a more heterogeneous pattern of enhancement within the tumor. Interestingly, while ADC values have often been shown to inversely correlate with ISUP grade [64,65], in our study, there were no ADC features present in the top 10 for any of the 3 imaging classifiers. In line with our findings, when using a multicenter MRI dataset, Bengtsson et al. [66] were not able to identify a correlation between ADC and ISUP grades, suggesting that the lack of harmonization in MRI parameters used to acquire and derive ADC maps might dilute the value of this feature in a clinical situation involving different field strengths, scanners, and acquisition parameters. This could potentially be an explanation for our results, which made use of a multicenter training dataset to identify the most important features. The highest-ranking imaging features suggest that T2W, DCE wash-ins, peak frames, as well as MRDI maps, contain relevant features for risk stratification. For the goal of the present study, the highest-ranking imaging features using the LR model were used to further investigate the relationship between imaging and transcriptomic data.

Our results established a significant relationship between imaging features and transcriptomic characteristics. A set of 31 imaging features was correlated with 33 transcriptomic features, yielding 5 significant correlations, 3 of which were of moderate strength (|r|≥ 0.5)—Figure 7. The strongest significant correlations were seen between the perfusion-based imaging feature—MRDI *A* median—and the activity of the transcription factors STAT6 (−0.64) and TFAP2A (−0.50). A higher-order T2W textural feature was significantly correlated to the transcription factor STAT6 (−0.58). The STAT6 expression plays a role in PCa cell proliferation and migration and was found to be positively correlated with the tumor size and high PCa histological grades [67]. TFAP2A is a well-characterized transcription factor that regulates proliferation and differentiation in mammalian cells [68]. Conversely, the loss of the AP2alpha protein expression has been strongly associated with the aggressiveness and progression of numerous types of cancers, including prostate cancer [69]. In addition, we observed a significant correlation between the MRDI *A* median and the gene signature Decipher PCA (−0.40), as well as between the pathway PI3K and DCE wash-in GLRLM GLV kurtosis (−0.04), which were below the set threshold of |r|≥ 0.5. In the study by Hectors et al. [35], a moderate positive correlation (r = 0.40) was previously reported between a T2w GLCM texture feature and the Decipher risk score. It is important to note that the way the Decipher score is derived in the current study differs from the Decipher risk scores typically reported in the literature, derived using clinical-grade whole-transcriptome Decipher assays. Here, the Decipher score exhibits a moderate correlation with the parameter MRDI *A*, derived from DCE-MRI. Microvessel density has been associated with recurrence and prognosis in patients with PCa [70], and angiogenesis is an essential step for tumor growth and the initiation of metastasis [71]. Oto et al. [72] observed a significant moderate correlation (r = 0.44) between microvessel density and the DCE-MRI parameter $k_{ep}$ in prostate cancer (which describes the back flux rate constant between the extracellular space and the plasma). Previous findings, thus, provide support to our result that MRDI *A*, which quantitatively describes perfusion and extravasation associated with angiogenesis, is related to the Decipher score, which grades tumor aggressiveness. The PI3K pathway is involved in cellular proliferation and growth, and an elevated PI3K signaling is considered a hallmark of cancer [73]. Thus, it would not be surprising to find a correlation between PI3K and DCE-MRI features, as this MRI sequence is used for oncological purposes, precisely to highlight angiogenesis processes. In our study, despite the significance, the correlation we observe between PI3K and the DCE wash-in GLRLM GLV kurtosis is close to zero in value. The strength of the significant correlations observed between imaging and genomic features is higher in our study when compared to the values described in a similar study by Hectors et al. [35]. The analysis of immune cell-type fractions was not addressed in this manuscript as they are not easily interpretable in this context. Single-cell RNA-seq is a strong unbiased method used to identify specific tumor cell subpopulations. It can reveal cellular heterogeneity and possibly identify distinct cell populations in the tumor microenvironment [74], and it is an interesting alternative to bulk RNA-seq data that would likely boost the performance of transcriptomic feature-based models. However, MRI is, by default, a bulk low-resolution method, and in this regard, bulk RNA-seq provides more suitable data to match it. Alternatively, spatial transcriptomics [75] retrieves spatially resolved maps of gene expression profiles and can investigate tumor heterogeneity, which can be (cor)related to the MR imaging domain.

The aim of this work was to identify candidate imaging features that can potentially represent relevant phenotypes for individualized patient treatment. Our strategy consisted of a machine learning framework to stratify patients based on their ISUP grade. Although relevant and typically investigated, the ISUP grade is not a complete depiction of the complexity of PCa, and a more suitable clinical outcome, such as 5-year biochemical recurrence-free or metastasis-free survival, would have been preferred. Nonetheless, datasets encompassing this level of detailed clinical information, which require long follow-up times in combination with MRIs (and genomic data), are virtually impossible to obtain, retrospectively. In addition, the use of prostatectomy specimens as the reference standard facilitated a more accurate localization of pathology-proven lesions on the MRI grid, but it also limited the sample size and potentially biased our study towards the subgroup of surgically treated PCa patients. Unlike other imaging modalities, MRI intensities have arbitrary units. This fact alone can significantly affect the absolute values of the derived radiomic features and, thus, their robustness. The robustness of radiomic features in a multi-center setting is a prerequisite for the successful translation of radiomic biomarkers in a clinical workflow [76]. In order to mitigate these issues, intensity normalization of non-quantitative images (T2W and DCE) was performed based on the prostate region prior to feature extraction. Alternative methods for image normalization and their impacts on radiomic feature robustness have been investigated [77]; histogram matching seems to be the most promising normalization scheme. A future prospective study would enable data harmonization and the validation of our findings in a larger cohort.

## 5. Conclusions

In this study, we identified radiomic features derived from mpMRI that are predictive of clinically significant PCa and are significantly correlated with some of the corresponding transcriptomic features. Our results suggest that mpMRI imaging features hold the potential to decode tumor phenotypes, paving the way for a non-invasive assessment of PCa and more individualized treatment regimens.

## Figures and Tables

**Figure 1 cancers-15-03074-f001:**
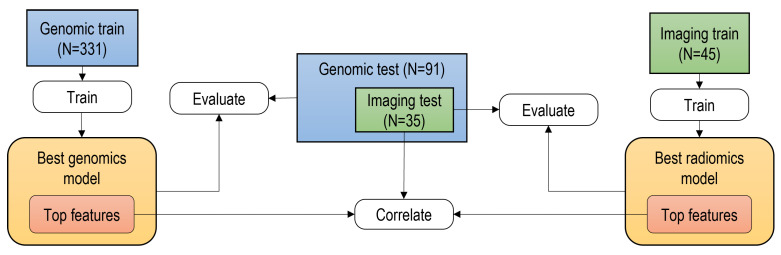
General framework and overview of the datasets used for the radiogenomics analysis. In this work, the genomic model was based on transcriptomic features, while the radiomics model was based on features extracted from mpMRI.

**Figure 2 cancers-15-03074-f002:**
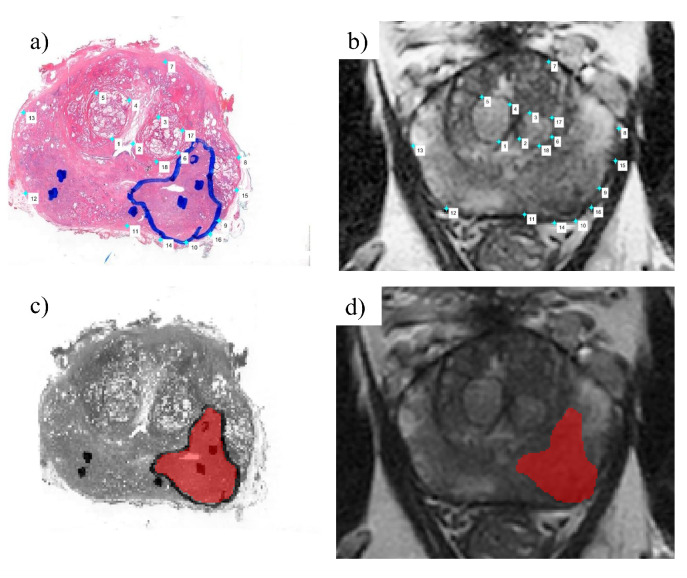
Illustration of the registration process and propagation of the tumor contour from the H&E slide to the matching T2W slice. (**a**) H&E-stained pathology slide with reference landmarks; (**b**) the matched T2W slice with reference landmarks; (**c**) deformably registered H&E slide; (**d**) H&E tumor contour propagated to the T2W slice.

**Figure 3 cancers-15-03074-f003:**
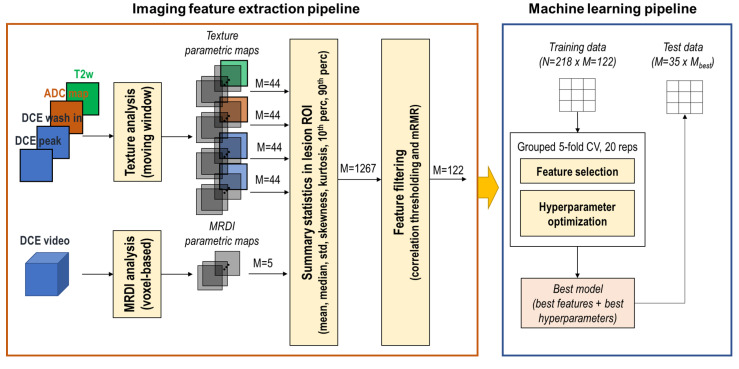
Overview of the imaging feature extraction and machine learning pipelines. *N* represents the number of samples, *M* represents the number of features.

**Figure 4 cancers-15-03074-f004:**
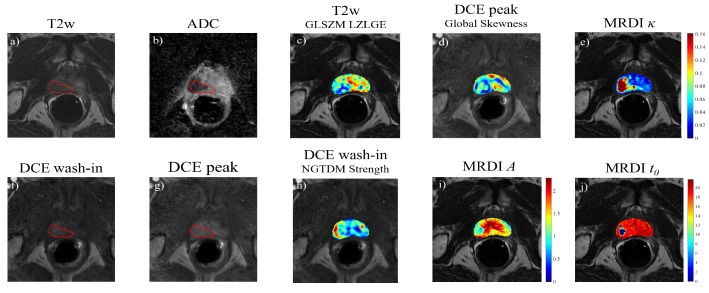
Example of mpMRI images (**a**) T2W, (**b**) ADC, (**f**) DCE wash-in frame, and (**g**) DCE peak frame), with corresponding texture feature maps (**c**) T2W GLSZM LZLGE, (**d**) DCE peak global skewness, and (**h**) DCE wash-in NGDTM strength and MRDI parametric maps, (**e**) κ, (**i**) *A*, and (**j**) $t_{0}$. The tumor contours are displayed in red.

**Figure 5 cancers-15-03074-f005:**
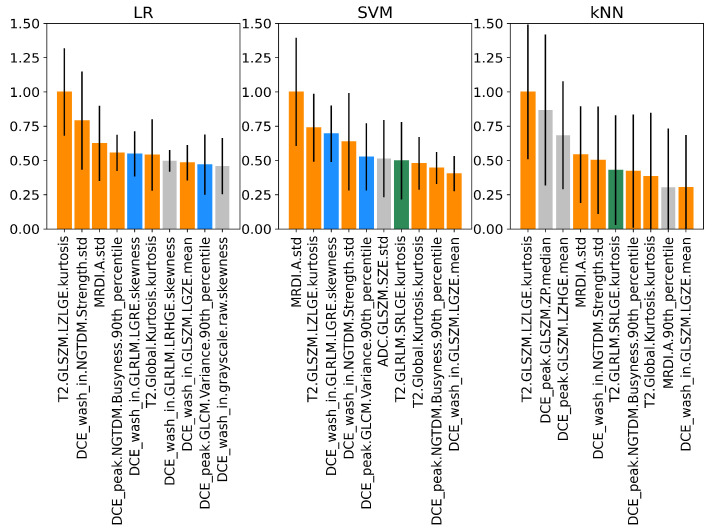
Normalized PFIs for LR, SVM, and kNN imaging classifiers on the training dataset. Features shared by all classifiers are shown in orange, the ones shared between LR and SVM are shown in blue, and the ones shared between SVM and kNN are shown in green. Features that are not shared are displayed in gray. The bars represent the standard deviations for the 10 random permutations.

**Figure 6 cancers-15-03074-f006:**
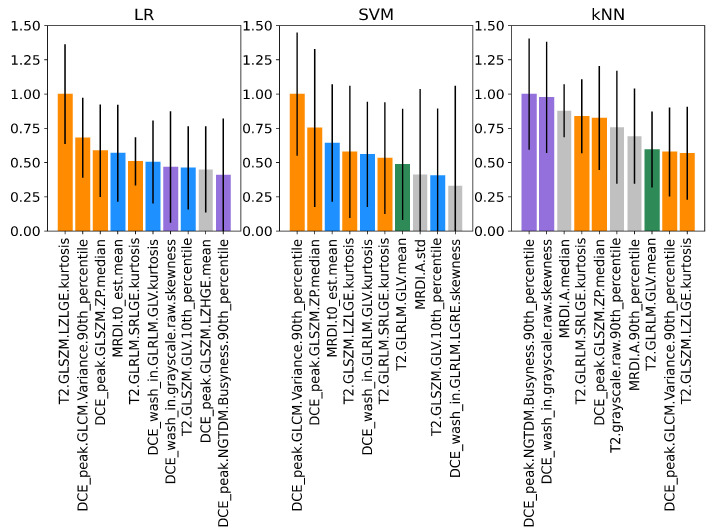
Normalized PFIs for LR, SVM, and kNN imaging classifiers on the test dataset. Features shared by all classifiers are shown in orange, the ones shared between LR and SVM are shown in blue, the ones shared between SVM and kNN are shown in green, and the ones shared by LR and kNN are shown in purple. Features that are not shared are displayed in gray. The bars represent the standard deviations for the 10 random permutations.

**Figure 7 cancers-15-03074-f007:**
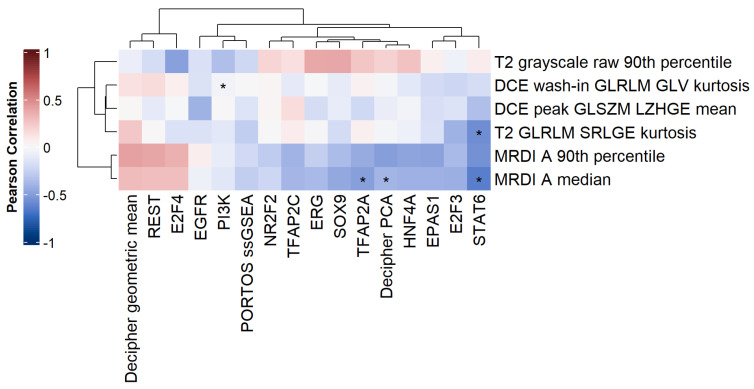
Heatmap of the Pearson correlation coefficient between imaging and transcriptomic features. Only coefficients with *p*-value ≤0.2 are shown. * *p* < 0.05.

**Table 1 cancers-15-03074-t001:** Overview of ISUP grades (index lesion) given in terms of the number of patients and respective percentages in each dataset (in parenthesis).

	Transcriptomics		Imaging	
	**Training**	**Test**	**Training**	**Test**
ISUP 1	64 (19.3%)	5 (5.5%)	11 (24.4%)	1 (2.9%)
ISUP 2	102 (30.8%)	29 (31.9%)	19 (42.2%)	10 (28.6%)
ISUP 3	78 (23.6%)	27 (29.7%)	9 (20.0%)	10 (28.6%)
ISUP 4	44 (13.3%)	15 (16.5%)	3 (6.7%)	8 (22.9%)
ISUP 5	43 (13.0%)	15 (16.5%)	3 (6.7%)	6 (17.1%)

**Table 2 cancers-15-03074-t002:** Overview of extracted texture features.

Texture Type	Features	Number of Features
Global	Variance, skewness, kurtosis	3
Gray-level co-occurrence matrix (GLCM)	Energy, contrast, correlation, homogeneity, variance, Sum average, entropy, dissimilarity, autocorrelation	9
Gray-level run-length matrix (GLRLM)	Short-run emphasis (SRE), long-run emphasis (LRE), gray-level non-uniformity (GLN), run-length non-uniformity (RLN), run percentage (RP), low gray-level run emphasis (LGRE), high gray-level run emphasis (HGRE), short-run low gray-level emphasis (SRLGE), short-run high gray-level emphasis (SRHGE), long-run low gray-level emphasis (LRLGE), long-run high gray-level emphasis (LRHGE), gray-level variance (GLV), run-length variance (RLV)	13
Gray-level size zone matrix (GLSZM)	Small zone emphasis (SZE), large zone emphasis (LZE), gray-level non-uniformity (GLN), zone-size non-uniformity (ZSN), zone percentage (ZP), low gray-level zone emphasis (LGZE), high gray-level zone emphasis (HGZE), small zone low gray-level emphasis (SZLGE), small zone high gray-level emphasis (SZHGE), large zone low gray-level emphasis (LZLGE), large zone high gray-level emphasis (LZHGE), gray-level variance (GLV), zone-size variance (ZSV)	13
Neighborhood gray-tone difference matrix (NGTDM)	Coarseness, contrast, business, Complexity, strength	5

**Table 3 cancers-15-03074-t003:** Description of the 33 selected transcriptomic features, consisting of 17 transcription factors, 10 signaling pathways and 6 gene signature scores.

Type of Transcriptomic Feature	Acronym	Full Name
Transcription factors	REST	RE1 silencing transcription factor
	FOXA2	Forkhead box protein A2
	TFDP1	Transcription factor Dp-1
	ERG	Erythroblast transformation specific (ETS) transcription factor ERG
	SOX9	SRY-box transcription factor 9
	TFAP2A	Transcription factor AP-2 alpha
	TFAP2C	Transcription factor AP-2 gamma
	HNF4A	Hepatocyte nuclear factor 4 alpha
	EPAS1	Endothelial PAS domain-containing protein 1
	E2F1	E2F transcription factor 1
	E2F2	E2F transcription factor 2
	E2F3	E2F transcription factor 3
	E2F4	E2F transcription factor 4
	BACH1	BTB and CNC homology 1
	STAT6	Signal transducer and activator of transcription 6
	AR	Androgen receptor
	NR2F2	Nuclear receptor subfamily 2 group F member 2
Pathways *	EGFR	Epidermal growth factor receptor—regulates growth, survival, migration, apoptosis, proliferation, and differentiation in mammalian cells.
	PI3K	Phosphatidylinositol 3-kinase—promotes growth and proliferation.
	Androgen	Involved in the growth and development of the male reproductive organs.
	Estrogen	Promotes the growth and development of the female reproductive organs.
	JAK-STAT	Janus kinases (JAKs), signal transducer and activator of transcription proteins (STATs)—involved in immunity, cell division, cell death, and tumor formation.
	MAPK	Mitogen-activated protein kinases (MAPKs)—integrates external signals and promotes cell growth and proliferation.
	TGFb	Transforming growth factor beta—involved in the development, homeostasis, and repair of most tissues.
	Trail	TNF-related apoptosis-inducing ligand—induces apoptosis.
	VEGF	Vascular endothelial growth factor—mediates angiogenesis, vascular permeability, and cell migration.
	WNT	Created from the names wingless and Int-1—regulates organ morphogenesis during development and tissue repair.
Signatures	Decipher geometric mean	Decipher signature, derived using the geometric mean
	Decipher PCA	Decipher signature, derived using PCA
	Decipher ssGSEA	Decipher signature, derived using single-sample gene set enrichment analysis
	PORTOS geometric mean	PORTOS signature, derived using the geometric mean
	PORTOS PCA	PORTOS signature, derived using PCA
	PORTOS ssGSEA	PORTOS signature, derived using single-sample gene set enrichment analysis

* The description provided for each pathway was derived from the PROGENy [49] pathway signatures GitHub repository.

**Table 4 cancers-15-03074-t004:** Overview of the optimal hyperparameters and optimal number of features for each classifier.

Classifier	Hyperparameter Description	Optimal Value
LR	Solver	Liblinear
	Penalty	l2
	C	0.02
	$N_{0}$	31
KNN	Weights	Uniform
	Metric	Minkowski
	K	28
	$N_{0}$	30
SVM	Kernel	Sigmoid
	γ	0.01
	C	0.95
	$N_{0}$	27

$N_{0}$—Optimal number of features; C—regularization parameter; K—number of neighbors; γ—Kernel coefficient.

**Table 5 cancers-15-03074-t005:** Performances of the three imaging classifiers on the imaging training and test datasets. The displayed values for the training dataset are the mean and standard deviations (in parenthesis) over the cross-validation scheme.

	Train			Test		
	**LR**	**KNN**	**SVM**	**LR**	**KNN**	**SVM**
ACC	0.72 (0.07)	0.67 (0.04)	0.75 (0.08)	0.71	0.34	0.69
bACC	0.72 (0.08)	0.55 (0.05)	0.74 (0.08)	0.74	0.52	0.72
Sensitivity	0.72 (0.14)	0.15 (0.11)	0.72 (0.16)	0.67	0.04	0.63
Specificity	0.72 (0.09)	0.96 (0.05)	0.76 (0.09)	0.82	1.00	0.82
AUC	0.82 (0.08)	0.72 (0.08)	0.83 (0.08)	0.74	0.52	0.72

ACC = accuracy; bACC = balanced accuracy; AUC = area under the curve.

## Data Availability

Data are not publicly available due to GDPR protection and data-sharing agreements.

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
