# Peer review of "Radiogenomics Analysis Linking Multiparametric MRI and Transcriptomics in Prostate Cancer"

_cancers, 2023, doi:10.3390/cancers15123074_

Round 1

Reviewer 1 Report

The manuscript "Radiogenomics analysis linking multiparametric MRI and transcriptomics in prostate cancer" contributes to understanding the relationship between imaging and transcriptomic features in aggressive prostate cancers (PCa). This study identified the mpMRI imaging and transcriptomic features which can be most predictive of aggressive PCa using machine learning and then analyzed their correlation.

In this work, a large set of texture features on all mpMRI imaging sequences was investigated using a moving-window strategy adopted to extract features at each imaging voxel, allowing the investigation of the feature spatial distribution and heterogeneity the tumor ROI. The authors performed magnetic resonance dispersion imaging (MRDI) on the DCE sequence to automatically estimate PK parameters related to microvascular perfusion, architecture, and permeability, without requiring an arterial-input function. Three machine learning models – logistic regression (LR), support vector machine (SVM), and k-nearest neighbors (kNN) were used to train and test of mpMRI dataset.

This study revealed correlations between the set of 31 imaging features and 33 transcriptomic features obtained on the same tumor samples: five significant correlations were found, of which three with moderate strength: (i) the strongest significant correlations were seen between a perfusion-based imaging feature - MRDI A median - and the activity of the TFs STAT6 (-0.64) and TFAP2A (-0.50); (ii) a higher-order T2W textural feature was significantly correlated to the activity of the TF STAT6 (-0.58), and (iii) loss of the AP2alpha protein expression (TFAP2A) has been strongly associated with aggressiveness and progression in PCa. The authors conclude that a combination of texture features extracted from T2W and DCE and perfusion-based pharmacokinetic features can be considered for the prediction of clinically significant PCa, with the pharmacokinetic MRDI A feature being the most correlated with the underlying transcriptomic information.

In my opinion, the present manuscript is an interesting, thorough, and accurate study.

The novelty and significance of this work are sufficiently substantiated in the Introduction and Discussion. These findings may be valuable for the diagnosis and treatment of patients with prostate cancers. The study design is appropriate, and most methods are adequately and fully described.

The description of most clinical and experimental procedures and results is detailed and complete.

The list of references is reasonably sufficient and includes all relevant publications. The conclusions are supported by the results. However, I have several comments and suggestions to improve the clarity of the data presentation.

1. It is unclear how many images (140+70 ROIs ?) from the training dataset were used to effectively train with the resulting model performance (AUC = 0.74) and optimize the used machine learning models. Please, clarify it in the Results.

2. The manuscript contains almost no description of methods for transcriptomic analysis and no data on the analysis of transcriptomic features, including selected 33 features used to determine correlations with image features. It is necessary to provide data with the results of all significant transcriptome characteristics ('132 functional features describing pathways and TFs activity') and selected 10 pathways and 17 transcription factors.

3. Figure 7, which demonstrates the heatmap of the Pearson correlation coefficient between imaging and transcriptomic feature, shows statistically significant correlations between MRDI A medium and Decipher PCA and between DCE wash-in GLRLM GLV kurtosis and PI3K. However, these correlations are not discussed in the Discussion. Nonetheless, it seems important to discuss these results too. It is also necessary to give the full name of all genes on this heatmap and mention Figure 7 in the Discussion.

4. Discussing the lack of correlations between image features and PSA expression would also be interesting.

5. In the Conclusions section, it is recommended to change the first sentence as it is worded grandiosely. Considering that only five significant correlations were found between transcriptomic and image features, the phrase "correlated with the underlying transcriptomic landscape" seems to be overstated.

Reviewer 2 Report

This article aims to combine diagnostic imaging with genomic information so that prostate cancer can be assessed in a non-invasive way, which has application value in diagnosis. In the study, the relationship between imaging and transcriptomic features are found in aggressive prostate cancer.

However, the biggest concern of this study is the quality of the dataset. In imaging training dataset, the microscopic analysis is performed by a pathologist to mark cancer regions, which depends a lot on the experience of a certain pathologist. If the regions are not very precise, the training model will be affected. Moreover, it is inefficient to divide regions by manual identification, leading the consequence of low number of imaging data, which will cause the data imbalance compared with transcriptomics data and affect machine learning model. Therefore, it is suggested to increase the number of imaging dataset to enhance credibility of this study, or instead, using machine learning way to delineate of tumor region, which is more efficient and can provide bigger dataset matching to transcriptomics.

As for transcriptomic dataset, this article uses bulk RNA seq data, and for future improvement, single cell RNA seq data maybe more suitable for improve the accuracy and sensitivity of training model.

Major issues:

1. The size of the data set is too small. As described in Section 3.1, in the training set of image data, only 30 patients with insignificant PCa and 15 patients with significant PCa were obtained, and whether the ROI data from the same patients increased the possibility of model overfitting. The total amount of data on the test set is only 35. Is the performance of the model guaranteed? The dataset should be increased and the performance of the models should be reevaluated.

2. Machine learning has been developed for a long time, and many classifier algorithms with excellent performance have been proposed. Why did the author choose LR, KNN and SVM as model classifiers instead of other classifier algorithms in this work?

3. The data in Table 4 of Section 3.3 shows that the AUC obtained by LR and SVM on the training set is 0.82 and 0.83. Although the AUC is low, it is acceptable. However, the AUC obtained by LR and SVM on the test set was only 0.74 and 0.72, which decreased by almost 0.1. The significant reduction of AUC indicates that there is a possibility of overfitting of the model on the training set, and reduces the reliability of the model.

Minor issues:

1. Please provide more details on the experimental part of RNA-seq.

Round 2

Reviewer 1 Report

The authors corrected the manuscript following the comments and answered all questions. Unclear points in the study have been clarified in the Methods and Results sections. The necessary explanations of some results has been added to the Discussion.

Reviewer 2 Report

All issues raised previously were either solved or responded. The manuscript can be accepted in present form.